# Expediting Large-Scale Vision Transformer for Dense Prediction without Fine-tuning

**Weicong Liang**[1][*]    **Yuhui Yuan**[4][*][†]    **Henghui Ding**[3]    **Xiao Luo**[2]
**Weihong Lin**[4]  **Ding Jia**[1]    **Zheng Zhang**[4]    **Chao Zhang**[1]    **Han Hu**[4]

[1]Key Laboratory of Machine Perception (MOE)
School of Intelligence Science and Technology, Peking University
[2]School of Mathematical Sciences, Peking University    [3]ETH Zurich
[4]Microsoft Research Asia

## Abstract

Vision transformers have recently achieved competitive results across various vision tasks but still suffer from heavy computation costs when processing a large number of tokens. Many advanced approaches have been developed to reduce the total number of tokens in large-scale vision transformers, especially for image classification tasks. Typically, they select a small group of essential tokens according to their relevance with the `[class]` token, then fine-tune the weights of the vision transformer. Such fine-tuning is less practical for dense prediction due to the much heavier computation and GPU memory cost than image classification.

In this paper, we focus on a more challenging problem, *i.e.*, accelerating large-scale vision transformers for dense prediction without any additional re-training or fine-tuning. In response to the fact that high-resolution representations are necessary for dense prediction, we present two non-parametric operators, a *token clustering layer* to decrease the number of tokens and a *token reconstruction layer* to increase the number of tokens. The following steps are performed to achieve this: (i) we use the token clustering layer to cluster the neighboring tokens together, resulting in low-resolution representations that maintain the spatial structures; (ii) we apply the following transformer layers only to these low-resolution representations or clustered tokens; and (iii) we use the token reconstruction layer to re-create the high-resolution representations from the refined low-resolution representations. The results obtained by our method are promising on five dense prediction tasks, including object detection, semantic segmentation, panoptic segmentation, instance segmentation, and depth estimation. Accordingly, our method accelerates $40\% \uparrow$ FPS and saves $30\% \downarrow$ GFLOPs of "Segmenter+ViT-L/16" while maintaining $99.5\%$ of the performance on ADE20K without fine-tuning the official weights.

## 1  Introduction

Transformer [66] has made significant progress across various challenging vision tasks since pioneering efforts such as DETR [4], Vision Transformer (ViT) [17], and Swin Transformer [46]. By removing the local inductive bias [18] from convolutional neural networks [27, 63, 59], vision transformers armed with global self-attention show superiority in scalability for large-scale models and billion-scale dataset [17, 82, 60], self-supervised learning [26, 74, 1], connecting vision and language [52, 33], etc. We can find from recent developments of SOTA approaches that

---

[*]Equal contribution.

[†]✉ yuhui.yuan@microsoft.com

vision transformers have dominated various leader-boards, including but not limited to image classification [72, 14, 16, 82], object detection [84, 45, 37], semantic segmentation [31, 11, 3], pose estimation [75], image generation [83], and depth estimation [41].

Although vision transformers have achieved more accurate predictions in many vision tasks, large-scale vision transformers are still burdened with heavy computational overhead, particularly when processing high-resolution inputs [19, 45], thus limiting their broader application to more resource-constrained applications and attracting efforts on re-designing light-weight vision transformer architectures [10, 50, 85]. In addition to this, several recent efforts have investigated how to decrease the model complexity and accelerate vision transformers, especially for image classification, and introduced various advanced approaches to accelerate vision transformers. Dynamic ViT [54] and EViT [43], for example, propose two different dynamic token sparsification frameworks to reduce the redundant tokens progressively and select the most informative tokens according to the scores predicted with an extra trained prediction module or their relevance with the [class] token. TokenLearner [57] learns to spatially attend over a subset of tokens and generates a set of clustered tokens adaptive to the input for video understanding tasks. Most of these token reduction approaches are carefully designed for image classification tasks and require fine-tuning or retraining. These approaches might not be suitable to tackle more challenging dense prediction tasks that need to process high-resolution input images, e.g., $1024 \times 1024$, thus, resulting in heavy computation and GPU memory cost brought. We also demonstrate in the supplemental material the superiority of our method over several representative methods on dense prediction tasks.

Rather than proposing a new lightweight architecture for dense prediction or token reduction scheme for only image classification, we focus on how to expedite well-trained large-scale vision transformers and use them for various dense prediction tasks without fine-tuning or re-training. Motivated by these two key observations including (i) the intermediate token representations of a well-trained vision transformer carry a heavy amount of local spatial redundancy and (ii) dense prediction tasks require high-resolution representations, we propose a simple yet effective scheme to convert the "high-resolution" path of the vision transformer to a "high-to-low-to-high resolution" path via two non-parametric layers including a token clustering layer and a token reconstruction layer. Our method can produce a wide range of more efficient models without requiring further fine-tuning or re-training. We apply our approach to expedite two main-stream vision transformer architectures, e.g., ViTs and Swin Transformers, for five challenging dense prediction tasks, including object detection, semantic segmentation, panoptic segmentation, instance segmentation, and depth estimation. We have achieved encouraging results across several evaluated benchmarks and Figure 1 illustrates some representative results on both semantic segmentation and depth estimation tasks.

## 2 Related work

**Pruning Convolutional Neural Networks.** Convolutional neural network pruning [2, 29, 69] is a task that involves removing the redundant parameters to reduce the model complexity without a significant performance drop. Pruning methods typically entail three steps: (i) training a large, over-parameterized model to convergence, (ii) pruning the trained large model according to a certain criterion, and (iii) fine-tuning the pruned model to regain the lost performance [48]. The key idea is to design an importance score function that is capable of pruning the less informative parameters. We follow [7] to categorize the existing methods into two main paths: (i) unstructured pruning (also named weight pruning) and (ii) structured pruning. Unstructured pruning methods explore the absolute value of each weight or the product of each weight and its gradient to estimate the importance scores. Structured pruning methods, such as layer-level pruning [70], filter-level pruning[47, 77], and image-level pruning [24, 64], removes the model sub-structures. Recent studies [5, 6, 30] further extend these pruning methods to vision transformer. Unlike the previous pruning methods, we explore how to expedite vision transformers for dense prediction tasks by carefully reducing & increasing the number of tokens without removing or modifying the parameters.

**Efficient Vision Transformer.** The success of vision transformers has incentivised many recent efforts [57, 61, 49, 28, 71, 54, 36, 23, 67, 9, 34, 43, 56] to exploit the spatial redundancies of intermediate token representations. For example, TokenLearner [57] learns to attend over a subset of tokens and generates a set of clustered tokens adaptive to the input. They empirically show that very few clustered tokens are sufficient for video understanding tasks. Token Pooling [49] exploits a nonuniform data-aware down-sampling operator based on K-Means or K-medoids to cluster similar

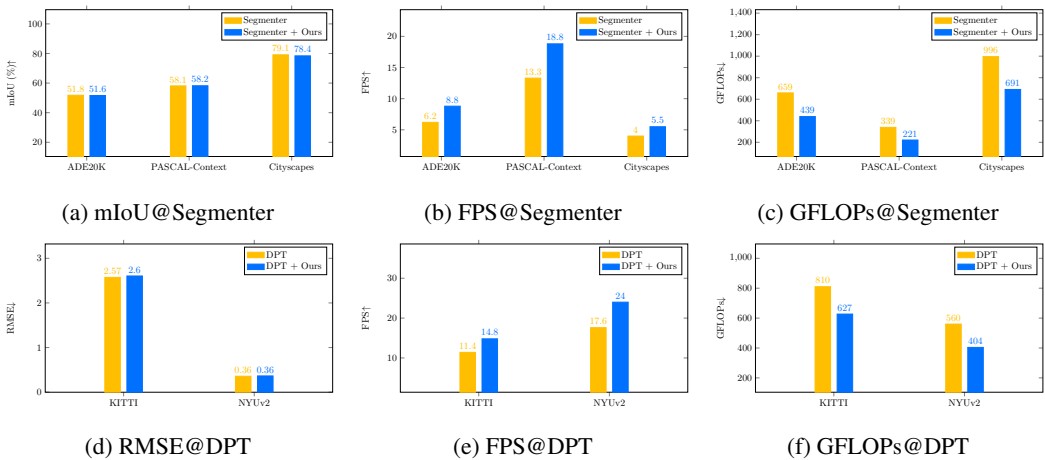

| (a) mIoU@Segmenter | (b) FPS@Segmenter | (c) GFLOPs@Segmenter |
|---|---|---|
| (d) RMSE@DPT | (e) FPS@DPT | (f) GFLOPs@DPT |

Figure 1: Illustrating the improvements of our approach: we report the results of applying our approach to Segmenter [62] for semantic segmentation and DPT [53] for depth estimation on the 1-st and 2-ed row respectively. Without any fine-tuning, our proposed method reduces the GFLOPs and accelerates the FPS significantly with a slight performance drop on both dense prediction tasks. ↑ and ↓ represent higher is better and lower is better respectively. Refer to Section 4 for more details.

tokens together to reduce the number of tokens while minimizing the reconstruction error. Dynamic ViT [54] observes that the accurate image recognition with vision transformers mainly depends on a subset of the most informative tokens, and hence it develops a dynamic token sparsification framework for pruning the redundant tokens dynamically based on the input. EViT (expediting vision transformers) [43] proposes to calculate the attentiveness of the [class] token with respect to each token and identify the top-$k$ attentive tokens according to the attentiveness score. Patch Merger [55] uses a learnable attention matrix to merge and combine together the redundant tokens, therefore creating a much more practical and cheaper model with only a slight performance drop. Refer to [65] for more details on efficient transformer architecture designs, such as Performer [12] and Reformer [35]. In contrast to these methods that require either retraining or fine-tuning the modified transformer architectures from scratch or the pre-trained weights, our approach can reuse the once-trained weights for free and produce lightweight models with a modest performance drop.

**Vision Transformer for Dense Prediction.** In the wake of success of the representative pyramid vision transformers [46, 68] for object detection and semantic segmentation, more and more efforts have explored different advanced vision transformer architecture designs [38, 8, 39, 40, 20, 79, 76, 86, 22, 42, 73, 78, 15, 81, 80, 25] suitable for various dense prediction tasks. For example, MViT [39] focuses more on multi-scale representation learning, while HRFormer [79] examines the benefits of combining multi-scale representation learning and high-resolution representation learning. Instead of designing a novel vision transformer architecture for dense prediction, we focus on how to accelerate a well-trained vision transformer while maintaining the prediction performance as much as possible.

**Our approach.** The contribution of our work lies in two main aspects: (i) we are the first to study how to accelerate state-of-the-art large-scale vision transformers for dense prediction tasks without fine-tuning (e.g., "Mask2Former + Swin-L" and "SwinV2-L + HTC++"). Besides, our approach also achieves much better accuracy and speedup trade-off when compared to the very recent ACT [1] which is based on a clustering attention scheme; (ii) our token clustering and reconstruction layers are capable of maintaining the semantic information encoded in the original high-resolution representations. This is the very most important factor to avoid fine-tuning. We design an effective combination of a token clustering function and a token reconstruction function to maximize the cosine similarity between the reconstructed high-resolution feature maps and the original ones without fine-tuning. The design of our token reconstruction layer is the key and not straightforward essentially. We also show that our token reconstruction layer can be used to adapt the very recent EViT [43] and DynamicViT [54] for dense prediction tasks in the supplementary.

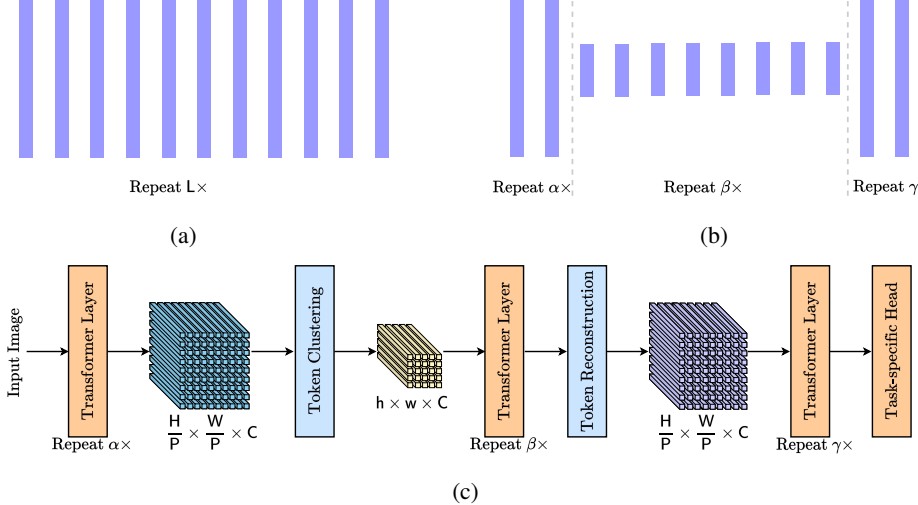

Figure 2: (a) Plain high-resolution vision transformer with L layers. (b) U-shape High-to-low-to-high-resolution vision transformer with $\alpha$, $\beta$, and $\gamma$ layers respectively ($L = \alpha + \beta + \gamma$). (c) Illustrating the details of using our approach to plain ViTs: we insert a token clustering layer and a token reconstruction layer into a trained vision transformer in order to decrease and then increase the spatial resolution, respectively. The weights of modules marked with ▢ are trained once based on the configuration of (a). The token clustering layer and token reconstruction layer are marked with ▢ are non-parametric, thus do not require any fine-tuning and can be included directly during evaluation.

## 3  Our Approach

**Preliminary.** The conventional Vision Transformer [17] first reshapes the input image $\mathbf{X} \in \mathbb{R}^{H \times W \times 3}$ into a sequence of flatten patches $\mathbf{X}_p \in \mathbb{R}^{N \times (P^2 3)}$, where $(P, P)$ represents the resolution of each patch, $(H, W)$ represents the resolution of the input image, $N = (H \times W)/P^2$ represents the number of resulting patches or tokens, *i.e.*, the input sequence length. The Vision Transformer consists of alternating layers of multi-head self-attention (MHSA) and feed-forward network (FFN) accompanied with layer norm (LN) and residual connections:

$$
\begin{aligned}
\mathbf{Z}_l^{'} &= \mathrm{MHSA}\left(\mathrm{LN}\left(\mathbf{Z}_{l-1}\right)\right) + \mathbf{Z}_{l-1}, \\
\mathbf{Z}_l &= \mathrm{FFN}\left(\mathrm{LN}\left(\mathbf{Z}_l^{'}\right)\right) + \mathbf{Z}_l^{'},
\end{aligned}
\tag{1}
$$

where $l \in \{1, \ldots, L\}$ represents the layer index, $\mathbf{Z}_l \in \mathbb{R}^{N \times C}$, and $\mathbf{Z}_0$ is based on $\mathbf{X}_p$. The computation cost $\mathcal{O}(LNC(N + C))$ mainly depends on the number of layers L, the number of tokens N, and the channel dimension C.

Despite the great success of transformer, its computation cost increases significantly when handling high-resolution representations, which are critical for dense prediction tasks. This paper attempts to resolve this issue by reducing the computation complexity during the inference stage, and presents a very simple solution for generating a large number of efficient vision transformer models directly from a single trained vision transformer, requiring no further training or fine-tuning.

We demonstrate how our approach could be applied to the existing standard Vision Transformer in Figure 2. The original Vision Transformer is modified using two non-parametric operations, namely a token clustering layer and a token reconstruction layer. The proposed token clustering layer is utilized to convert the high-resolution representations to low-resolution representations by clustering the locally semantically similar tokens. Then, we apply the following transformer layers on the low-resolution representations, which greatly accelerates the inference speed and saves computation resources. Last, a token reconstruction layer is proposed to reconstruct the feature representations back to high-resolution.

**Token Clustering Layer.** We construct the token clustering layer following the improved SLIC scheme [32], which performs local k-means clustering as follows:

-*Initial superpixel center*: We apply adaptive average pooling (AAP) over the high-resolution representations from the $\alpha$-th layer to compute the $\mathsf{h} \times \mathsf{w}$ initial cluster center representations:

$$\mathbf{S}_\alpha = \mathrm{AAP}(\mathbf{Z}_\alpha, (\mathsf{h} \times \mathsf{w})), \tag{2}$$

where $\mathbf{S}_\alpha \in \mathbb{R}^{\mathsf{hw} \times \mathsf{C}}$, $\mathbf{Z}_\alpha \in \mathbb{R}^{\mathsf{N} \times \mathsf{C}}$, and $\mathsf{hw} \ll \mathsf{N}$.

-*Iterative local clustering*: (i) Expectation step: compute the normalized similarity between each pixel $p$ and the surrounding superpixel $i$ (we only consider the neighboring $\lambda$ positions), (ii) Maximization step: compute the new superpixel centers:

$$\mathbf{Q}_{p,i} = \frac{\exp\left(-||\mathbf{Z}_{\alpha,\mathsf{p}} - \mathbf{S}_{\alpha,\mathsf{i}}||^2/\tau\right)}{\sum_{j=1}^{\lambda} \exp\left(-||\mathbf{Z}_{\alpha,\mathsf{p}} - \mathbf{S}_{\alpha,\mathsf{j}}||^2/\tau\right)}, \quad \mathbf{S}_{\alpha,i} = \sum_{p=1}^{\mathsf{N}} \mathbf{Q}_{p,i}\mathbf{Z}_{\alpha,p}, \tag{3}$$

where we iterate the above Expectation step and Maximization step for $\kappa$ times, $\tau$ is a temperature hyper-parameter, and $i \in \{1, 2, \cdots, \lambda\}$. We apply the following $\beta$ transformer layers on $\mathbf{S}_\alpha$ instead of $\mathbf{Z}_\alpha$, thus results in $\mathbf{S}_{\alpha+\beta}$ and decreases the computation cost significantly.

**Token Reconstruction Layer.** We implement the token reconstruction layer by exploiting the relations between the high-resolution representations and the low-resolution clustered representations:

$$\mathbf{Z}_{\alpha+\beta,p} = \sum_{\mathbf{S}_{\alpha,i} \in \text{k-NN}(\mathbf{Z}_{\alpha,p})} \frac{\exp\left(-||\mathbf{Z}_{\alpha,p} - \mathbf{S}_{\alpha,i}||^2/\tau\right)}{\sum_{\mathbf{S}_{\alpha,j} \in \text{k-NN}(\mathbf{Z}_{\alpha,p})} \exp\left(-||\mathbf{Z}_{\alpha,p} - \mathbf{S}_{\alpha,j}||^2/\tau\right)} \mathbf{S}_{\alpha+\beta,i}, \tag{4}$$

where $\tau$ is the same temperature hyper-parameter as in Equation 3. k-NN$(\mathbf{Z}_{\alpha,p})$ represents a set of the k nearest, a.k.a, most similar, superpixel representations for $\mathbf{Z}_{\alpha,i}$. We empirically find that choosing the same neighboring positions as in Equation 3 achieves close performance as the k-NN scheme while being more easy to implementation.

In summary, we estimate their semantic relations based on the representations before refinement with the following $\beta$ transformer layers and then reconstruct the high-resolution representations from the refined low-resolution clustered representations accordingly.

Finally, we apply the remained $\gamma$ transformer layers to the reconstructed high-resolution features and the task-specific head on the refined high-resolution features to predict the target results such as semantic segmentation maps or monocular depth maps.

**Extension to Swin Transformer.** We further introduce the *window token clustering layer* and *window token reconstruction layer*, which are suitable for Swin Transformer [45, 46]. Figure 3 illustrates an example usage of the proposed window token clustering layer and window token reconstruction layer. We first cluster the $\mathsf{K} \times \mathsf{K}$ window tokens into $\mathsf{k} \times \mathsf{k}$ window tokens and then reconstruct $\mathsf{K} \times \mathsf{K}$ window tokens according to the refined $\mathsf{k} \times \mathsf{k}$ window tokens. We apply the swin transformer layer equipped with smaller window size $\mathsf{k} \times \mathsf{k}$ on the clustered representations, where we need to bi-linear interpolate the pre-trained weights of relative position embedding table from $(2\mathsf{K} - 1) \times (2\mathsf{K} - 1)$ to $(2\mathsf{k} - 1) \times (2\mathsf{k} - 1)$ when processing the clustered representations. In summary, we can improve the efficiency of Swin Transformer by injecting the window token clustering layer and the window token reconstruction layer into the backbones seamlessly without fine-tuning the model weights.

**Why our approach can avoid fine-tuning?** The reasons include the following two aspects: (i) our token clustering/reconstruction layers are non-parametric, thus avoiding retraining any additional parameters, (ii) the reconstructed high-resolution representations maintain high semantic similarity with the original high-resolution representations. We take Segmenter+ViT-L/16 (on ADEK, $\alpha$=10) as an example and analyze the semantic similarity between the reconstructed high-resolution feature (with our approach) and the original high-resolution feature (with the original ViT-L/16) in Table 1. Accordingly, we can see that the cosine similarities are consistently high across different transformer layers between the reconstructed high-resolution feature (with our approach) and the original high-resolution feature. In other words, our approach well maintains the semantic information carried in the original high-resolution feature maps and thus is capable of avoiding fine-tuning.

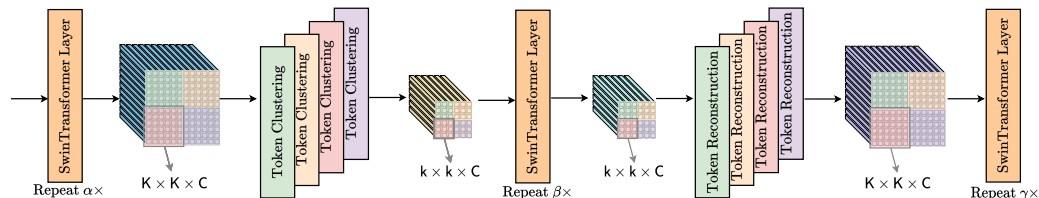

Figure 3: Illustrating the details of using our approach for Swin Transformer [45, 46]: we apply four groups of token clustering layer and token reconstruction layer within the four non-overlapped windows marked with ■, ■, ■, and ■ respectively, in the example referred to as *window token clustering layer* and *window token reconstruction layer*. We apply the intermediate swin transformer layers equipped with window size k × k) on the clustered window tokens. Window sizes before and after token clustering layer are K × K and k × k, and vice versa, for the token reconstruction layer.

Table 1: Ablation of the cosine similarity between the reconstructed high-resolution feature maps and the original high-resolution feature maps, where $\mathbf{Z}_{\alpha+\beta}^{\text{original}}$ and $\mathbf{Z}_{\alpha+\beta}$ represents the original and reconstructed high-resolution feature maps, respectively.

| $\alpha+\beta$ | 12 | 14 | 16 | 18 | 20 | 22 | 24 |
|---|---|---|---|---|---|---|---|
| $\cos(\mathbf{Z}_{\alpha+\beta}, \mathbf{Z}_{\alpha+\beta}^{\text{original}})$ | 0.94 | 0.95 | 0.96 | 0.96 | 0.96 | 0.96 | 0.96 |

# 4 Experiment

We verify the effectiveness of our method across five challenging dense prediction tasks, including object detection, semantic segmentation, instance segmentation, panoptic segmentation, and monocular depth estimation. We carefully choose the advanced SOTA methods that build the framework based on either the plain ViTs [17] or the Swin Transformers [45, 46]. We can integrate the proposed token clustering layer and token reconstruction layer seamlessly with the provided official trained checkpoints with no further fine-tuning required. More experimental details are illustrated as follows.

## 4.1 Datasets

**COCO** [44]. This dataset consists of 123K images with 896K annotated bounding boxes belonging to 80 thing classes and 53 stuff classes, where the `train` set contains 118K images and the `val` set contains 5K images. We report the object detection performance of SwinV2 + HTC++ [45] and the instance/panoptic segmentation performance of Mask2Former [11] on the `val` set.

**ADE**20**K** [87]. This dataset contains challenging scenes with fine-grained labels and is one of the most challenging semantic segmentation datasets. The `train` set contains $20,210$ images with $150$ semantic classes. The `val` set contains $2,000$ images. We report the segmentation results with Segmenter [62] on the `val` set.

**PASCAL-Context** [51]. This dataset consists of 59 semantic classes plus a background class, where the `train` set contains $4,996$ images with and the `val` set contains $5,104$ images. We report the segmentation results with Segmenter [62] on the `val` set.

**Cityscapes** [13]. This dataset is an urban scene understanding dataset with 30 classes while only 19 classes are used for parsing evaluation. The `train` set and `val` set contains $2,975$ and $500$ images respectively. We report the segmentation results with Segmenter [62] on the `val` set.

**KITTI** [21]. This dataset provides stereo, optical flow, visual odometry (SLAM), and 3D object detection of outdoor scenes captured by equipment mounted on a moving vehicle. We choose DPT [53] as our baseline to conduct experiments on the monocular depth prediction tasks, which consists of around 26K images for `train` set and 698 images for `val` set, where only 653 images have the ground-truth depth maps and the image resolution is of $1,241 \times 376$.

**NYUv**2 [58]. This dataset consists of $1,449$ RGBD images with resolution of $640 \times 480$, which captures $464$ diverse indoor scenes and contains rich detailed dense annotations such as surface normals, segmentation maps, depth, 3D planes, and so on. We report the depth prediction results of DPT [53] evaluated on 655 `val` images.

Table 2: Influence of the hyper-parameters of token clustering/reconstruction layer.

| Parameter | $\lambda$ | | | $\kappa$ | | | | $\tau$ | | | | k | | |
|---|---|---|---|---|---|---|---|---|---|---|---|---|---|---|
| | $3 \times 3$ | $5 \times 5$ | $7 \times 7$ | 5 | 10 | 15 | 20 | 10 | 25 | 50 | 75 | 10 | 20 | 50 |
| GFLOPs | 438.6 | 438.9 | 439.6 | 438.9 | 439.6 | 440.2 | 440.8 | | 438.9 | | | 438.9 | 438.9 | 439.0 |
| mIoU | 51.46 | 51.56 | 51.55 | 51.56 | 51.59 | 51.62 | 51.62 | 51.18 | 51.35 | 51.56 | 51.07 | 51.34 | 51.56 | 51.56 |

Table 3: Influence of the cluster size h × w.

| Cluster size | $8 \times 8$ | $12 \times 12$ | $16 \times 16$ | $20 \times 20$ | $24 \times 24$ | $28 \times 28$ | $40 \times 40$ (baseline) |
|---|---|---|---|---|---|---|---|
| GFLOPs | 274.0 | 290.9 | 315.1 | 347.2 | 388.2 | 438.9 | 659.0 |
| mIoU | 32.13 | 44.01 | 48.21 | 50.17 | 51.32 | 51.56 | 51.82 |

Table 4: Comparison with adaptive average pooling(AAP).

| Cluster size | $20 \times 20$ | $28 \times 28$ |
|---|---|---|
| AAP | 46.45 | 46.54 |
| Ours | 50.17 | 51.56 |

Table 5: Comparison with bi-linear upsample.

| Cluster size | $20 \times 20$ | $28 \times 28$ |
|---|---|---|
| Bi-linear | 44.68 | 44.74 |
| Ours | 50.17 | 51.56 |

Table 6: Combination with lighter vision transformer backbone.

| Cluster size | mIoU | GFLOPs |
|---|---|---|
| Segmenter+ViT-B/16 | 48.48 | 124.7 |
| Segmenter+ViT-B/16+Ours | 48.40 | 91.9 |

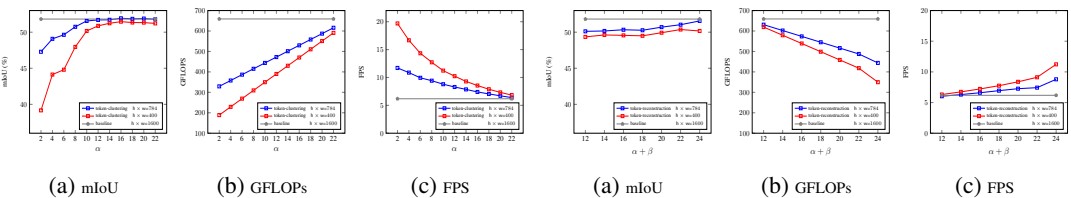

(a) mIoU    (b) GFLOPs    (c) FPS    (a) mIoU    (b) GFLOPs    (c) FPS

Figure 4: Influence of the inserted position $\alpha$ of token clustering layer.

Figure 5: Influence of the inserted position $\alpha + \beta$ of token reconstruction layer.

## 4.2 Evaluation Metrics

We report the numbers of AP (average precision), mask AP (mask average precision), PQ (panoptic quality), mIoU (mean intersection-over-union), and RMSE (root mean squared error) across object detection, instance segmentation, panoptic segmentation, semantic segmentation, and depth estimation tasks respectively. Since the dense prediction tasks care less about the throughput used in image classification tasks [43], we report FPS to measure the latency and the number of GFLOPs to measure the model complexity during evaluation. FPS is tested on a single V100 GPU with Pytorch 1.10 and CUDA 10.2 by default. More details are provided in the supplementary material.

## 4.3 Ablation Study Experiments

We conduct the following ablation experiments on ADE20K semantic segmentation benchmark with the official checkpoints of Segmenter+ViT-L/16 [62][3] by default if not specified.

**Hyper-parameters of token clustering/reconstruction layer**. We first study the influence of the hyper-parameters associated with the token clustering layer, i.e., the number of neighboring pixels $\lambda$ used in Equation 3, the number of EM iterations $\kappa$, and the choice of the temperature $\tau$ in Table 2. According to the results, we can see that our method is relatively less sensitive to the choice of both $\lambda$ and $\kappa$ compared to $\tau$. In summary, we choose $\lambda$ as $5 \times 5$, $\kappa$ as 5, and $\tau$ as 50 considering both performance and efficiency. Next, we also study the influence of the hyper-parameters within the token clustering layer, i.e., the number of nearest neighbors k within k-NN. We do not observe obvious differences and thus set k as 20. More details are provided in the supplementary material.

**Influence of cluster size choices**. We study the influence of different cluster size h × w choices based on input feature map of size $\frac{H}{P} \times \frac{W}{P} = 40 \times 40$ (N = 1,600)[4] in Table 3. According to the results, we can see that choosing too small cluster sizes significantly harms the dense prediction performance, and setting h × w as $28 \times 28$ achieves the better trade-off between performance drop and model complexity. Therefore, we choose $28 \times 28$ on Segmenter+ViT-L/16 by default. We also

---

[3] https://github.com/rstrudel/segmenter#ade20k, MIT License
[4] We choose $H \times W = 640 \times 640$ and $P = 16$, thus, $\frac{H}{P} \times \frac{W}{P} = 40 \times 40$ or $N = 1,600$, on ADE20K.

empirically find that selecting the cluster size $h \times w$ around $N/4 \sim N/2$ performs better on most of the other experiments. We conduct the following ablation experiments under two typical settings, including $20 \times 20$ ($\sim N/4$) and $28 \times 28$ ($\sim N/2$).

**Comparison with adaptive average pooling and bi-linear upsample**. We report the comparison results between our proposed token clustering/reconstruction scheme and adaptive average pooling/bi-linear upsample scheme in Table 4 and Table 5 under two cluster size settings respectively. We choose to compare with adaptive average pooling and bi-linear upsample instead of strided convolution or deconvolution as the previous ones are non-parametric and the later ones require re-training or fine-tuning, which are not the focus of this work. Specifically, we keep the inserted position choices the same and only replace the token cluster or token reconstruction layer with adaptive average pooling or bi-linear upsampling under the same cluster size choices. According to the results, we can see that our proposed token clustering and token reconstruction consistently outperform adaptive average pooling and bi-linear upsampling under different cluster size choices.

**Influence of inserted position of token clustering/reconstruction layer**. We investigate the influence of the inserted position of both token clustering layer and token reconstruction layer and summarize the detailed results in Figure 4 and Figure 5 under two different cluster size choices. According to the results shown in Figure 4, our method achieves better performance when choosing $\alpha$ larger than 10, therefore, we choose $\alpha = 10$ as it achieves a better trade-off between model complexity and segmentation accuracy. Then we study the influence of the inserted positions of the token reconstruction layer by fixing $\alpha = 10$. According to Figure 5, we can see that our method achieves the best performance when setting $\alpha + \beta = 24$, in other words, we insert the token reconstruction layer after the last transformer layer of ViT-L/16. We choose $\alpha = 10$, $\alpha + \beta = 24$, and $\gamma = 0$ for all ablation experiments on ADE20K by default if not specified.

**Combination with lighter vision transformer architecture**. We report the results of applying our method to lighter vision transformer backbones such as ViT-B/16, in Table 6. Our approach consistently improves the efficiency of Segmenter+ViT-B/16 at the cost of a slight performance drop without fine-tuning. Specifically speaking, our approach saves more than $26\% \downarrow$ GFLOPs of a trained "Segmenter+ViT-B/16" with only a slight performance drop from $48.48\%$ to $48.40\%$, which verifies our method also generalizes to lighter vision transformer architectures.

**Comparison with uniform downsampling**. We compare our method with the simple uniform downsampling scheme, which directly downsamples the input image into a lower resolution. Figure 6 summarizes the detailed comparison results. For example, on ADE20K, we downsample the input resolution from $640 \times 640$ to smaller resolutions (*e.g.*, $592 \times 592$, $576 \times 576$, $560 \times 560$, and $544 \times 544$) and report their performance and GFLOPs in Figure 6. We also plot the results with our method and we can see that our method consistently outperforms uniform sampling on both ADE20K and PASCAL-Context under multiple different GFLOPs budgets.

## 4.4 Object Detection

We use the recent SOTA object detection framework SwinV2-L + HTC++ [45] as our baseline. We summarize the results of combining our method with SwinV2-L + HTC++ on COCO object detection and instance segmentation tasks in Figure 7.

**Implementation details.** The original SwinV2-L consists of $\{2,2,18,2\}$ shifted window transformer blocks across the four stages. We only apply our method to the 3-rd stage with 18 blocks considering it dominates the computation overhead, which we also follow in the "Mask2Former + Swin-L" experiments. We insert the window token clustering/reconstruction layer after the 8-th/18-th block within the 3-rd stage, which are based on the official checkpoints [5] of SwinV2-L + HTC++. In other words, we set $\alpha = 12$ and $\alpha + \beta = 22$ for SwinV2-L. The default window size is $K \times K = 32 \times 32$ and we set the clustered window size as $k \times k = 23 \times 23$. We choose the values of other hyperparameters following the ablation experiments. According to the results summarized in Figure 7, compared to SwinV2-L + HTC++, our method improves the FPS by $21\% \uparrow$ and saves the GFLOPs by nearly $20\% \downarrow$ while maintaining around $98\%$ of object detection & instance segmentation performance.

---

[5] https://github.com/microsoft/Swin-Transformer, MIT License

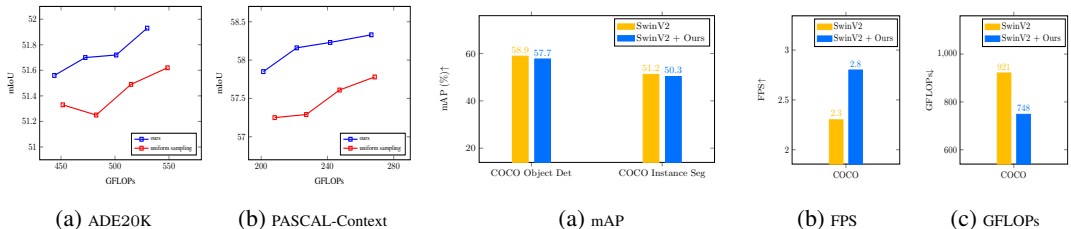

(a) ADE20K    (b) PASCAL-Context

Figure 6: Comparison to uniform downsampling on ADE20K and PASCAL-Context with Segmenter+ViT-L/16.

(a) mAP    (b) FPS    (c) GFLOPs

Figure 7: Illustrating the improvements of our approach on object detection task with SwinV2-L + HTC++. ↑ and ↓ represent higher is better and lower is better respectively.

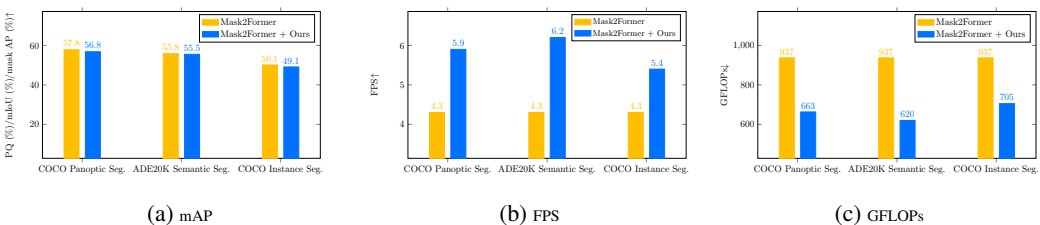

(a) mAP    (b) FPS    (c) GFLOPs

Figure 8: Illustrating the improvements of our approach on panoptic segmentation, semantic segmentation, and instance segmentation tasks with Mask2Former+Swin-L.

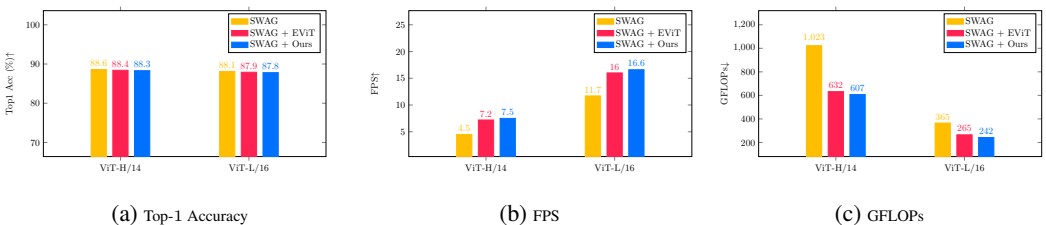

(a) Top-1 Accuracy    (b) FPS    (c) GFLOPs

Figure 9: Illustrating the improvements of our approach and the comparisons with EViT [43] on ImageNet classification tasks with SWAG+ViT-H/14 and SWAG+ViT-L/16.

## 4.5 Semantic/Instance/Panoptic Segmentation

We first apply our method to a plain ViT-based segmentation framework Segmenter [62] and illustrate the semantic segmentation results across three benchmarks including ADE20K, PASCAL-Context, and Cityscapes on the first row of Figure 1. Then, we apply our method to a very recent framework Mask2Former [11] that is based on Swin Transformer and summarize the semantic segmentation, instance segmentation, and panoptic segmentation results on COCO in Figure 8.

**Implementation details.** The original ViT-L first splits an image into a sequence of image patches of size $16 \times 16$ and applies a patch embedding layer to increase the channel dimensions to $1024$, then applies $24$ consecutive transformer encoder layers for representation learning. To apply our method to the ViT-L backbone of Segmenter, we use the official checkpoints [6] of "Segmenter + ViT-L/16" and insert the token clustering layers and token reconstruction layer into the ViT-L/16 backbone without fine-tuning.

For the Mask2Former built on Swin-L with window size as $12 \times 12$, we use the official checkpoints [7] of "Mask2Former + Swin-L" and insert the window token clustering layer and the window token reconstruction layer into the empirically chosen positions, which first cluster $12 \times 12$ tokens into $8 \times 8$ tokens and then reconstruct $12 \times 12$ tokens within each window. Figure 8 summarizes the detailed comparison results. Accordingly, we can see that our method significantly improves the FPS by more than $35\%$ ↑ with a slight performance drop on COCO panoptic segmentation task.

---

[6] https://github.com/rstrudel/segmenter#model-zoo, MIT License

[7] https://github.com/facebookresearch/Mask2Former/blob/main/MODEL_ZOO.md, CC-BY-NC $4.0$

Table 7: Depth estimation results based on DPT [53] with ResNet-50+ViT-B/16.

| Dataset | Method | GFLOPs | FPS | $\delta{>}1.25$ | $\delta{>}1.25^2$ | $\delta{>}1.25^3$ | AbsRel | SqRel | RMSE | RMSElog | SILog | log10 |
|---------|--------|--------|-----|-------|-------|-------|--------|-------|------|---------|-------|-------|
| KITTI | DPT | 810 | 11.38 | 0.959 | 0.995 | 0.999 | 0.062 | 0.222 | 2.573 | 0.092 | 8.282 | 0.027 |
| | DPT+Ours | 627 | 14.75 | 0.958 | 0.995 | 0.999 | 0.062 | 0.226 | 2.597 | 0.093 | 8.341 | 0.027 |
| NYUv2 | DPT | 560 | 17.58 | 0.904 | 0.988 | 0.998 | 0.110 | 0.054 | 0.357 | 0.129 | 9.522 | 0.045 |
| | DPT+Ours | 404 | 24.03 | 0.900 | 0.987 | 0.998 | 0.113 | 0.056 | 0.363 | 0.132 | 9.532 | 0.046 |

## 4.6 Monocular Depth Estimation

To verify the generalization of our approach, we apply our method to depth estimation tasks that measure the distance of each pixel relative to the camera. We choose the DPT (Dense Prediction Transformer) [53] that builds on the hybrid vision transformer, i.e., R50+ViT-B/16, following [17].

**Implementation details.** The original R50+ViT-B/16 [8] consists of a ResNet50 followed by a ViT-B/16, where the ViT-B/16 consists of 12 transformer encoder layers that process $16\times$ downsampled representations. We insert the token clustering layer & token reconstruction layer into ViT-B/16 and summarize the results on both KITTI and NYUv2 on the second row of Figure 1. We also report their detailed depth estimation results in Table 7, where we can see that our method accelerates DPT by nearly 30%/37% ↑ on KITTI/NYUv2, respectively.

## 4.7 ImageNet-1K Classification

Finally, we apply our method to the ImageNet-1K classification task and compare our method with a very recent SOTA method EViT [43]. The key idea of EViT is to identify and only keep the top-$k$ tokens according to their attention scores relative to the [class] token. We empirically find that applying EViT for dense prediction tasks directly suffers from significant performance drops. More details are illustrated in the supplementary material.

**Implementation details.** We choose the recent SWAG [60] as our baseline, which exploits 3.6 billion weakly labeled images associated with around 27K categories (or hashtags) to pre-train the large-scale vision transformer models, i.e., ViT-L/16 and ViT-H/14. According to their official implementations [9], SWAG + ViT-H/14 and SWAG + ViT-L/16 achieve 88.55% and 88.07% top-1 accuracy on ImaegNet-1K respectively. We apply our approach and EViT to both baselines and summarize the comparison results in Figure 9. According to the results, our method achieves comparable results as EViT while being more efficient, which further verifies that our method also generalizes to the image classification tasks without fine-tuning.

## 5 Conclusion

In this paper, we present a simple and effective mechanism to improve the efficiency of large-scale vision transformer models for dense prediction tasks. In light of the relatively high costs associated with re-training or fine-tuning large vision transformer models on various dense prediction tasks, our study provides a very lightweight method for expediting the inference process while requiring no additional fine-tuning. We hope our work could inspire further research efforts into exploring how to accelerate large-scale vision transformers for dense prediction tasks without fine-tuning.

## Acknowledgement

This work is partially supported by the National Nature Science Foundation of China under Grant 62071013 and 61671027, and National Key R&D Program of China under Grant 2018AAA0100300.

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
