# Supplementary for Expediting Large-Scale Vision Transformer for Dense Prediction without Fine-tuning

**Weicong Liang**[1][*]    **Yuhui Yuan**[4][*][†]    **Henghui Ding**[3]    **Xiao Luo**[2]
**Weihong Lin**[4]  **Ding Jia**[1]    **Zheng Zhang**[4]    **Chao Zhang**[1]    **Han Hu**[4]
[1]Key Laboratory of Machine Perception (MOE)
School of Intelligence Science and Technology, Peking University
[2]School of Mathematical Sciences, Peking University    [3]ETH Zurich
[4]Microsoft Research Asia

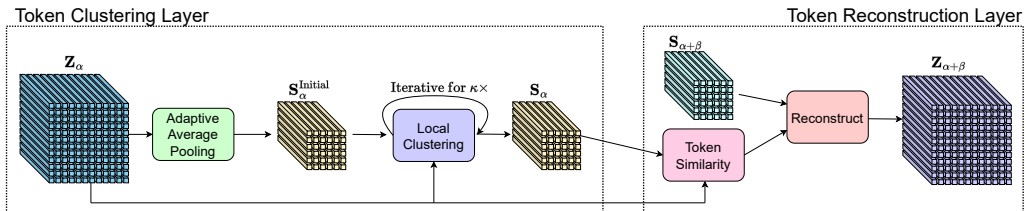

Figure 1: Illustrating more details of our approach. The token clustering layer consists of an adaptive average pooling block (for initializing the cluster centers) and an iterative local clustering block (for performing the k-means clustering). The token reconstruction layer consists of a token similarity estimation block (for estimating the reconstruction relation matrix) and a reconstruction block (for reconstructing the high-resolution representations). $\mathbf{Z}_\alpha$ represents the original high-resolution representations after $\alpha$-th transformer layer. $\mathbf{S}_\alpha$ represents the clustered low-resolution representations by token clustering layer. $\mathbf{S}_{\alpha+\beta}$ represents the refined clustered low-resolution representations after additional $\beta$ transformer layers. $\mathbf{Z}_{\alpha+\beta}$ represents the reconstructed high-resolution representations from $\mathbf{S}_{\alpha+\beta}$ by using the token reconstruction layer.

## A. Illustrating More Details of Our Approach

We first illustrate the overall details of our token clustering layer and token reconstruction layer in Figure 1. We then present the example implementation of token clustering layer and token reconstruction layer based on PyTorch in Listing 1 and Listing 2, respectively.

## B. More Hyper-parameter Details

We summarize the detailed hyper-parameter settings for the dense prediction methods based on plain ViTs and Swin Transformers in Table 1 and Table 2, respectively.

Table 1 summarizes the hyper-parameters, including the inserted positions $\alpha$ & $\alpha+\beta$ of token clustering layer & token reconstruction layer, the number of remaining transformer layers after the token reconstruction layer $\gamma$, the total number of transformer layers L, the number of tokens before clustering $\frac{H}{P} \times \frac{W}{P}$, the number of tokens after clustering $h \times w$, the number of neighboring pixels $\lambda$, the number of EM iterations $\kappa$, the temperature value $\tau$, and the number of nearest neighbors k, for Segmenter, DPT, and SWAG.

---

[*]Equal contribution.

[†]✉ yuhui.yuan@microsoft.com

36th Conference on Neural Information Processing Systems (NeurIPS 2022).

```
1  def token_clustering_layer(features, cluster_features_shape, num_iters, tau):
2      # args:
3      #       features: shape [B, C, H, W]
4      #       cluster_features_shape: [h, w]
5      #       num_iters: num of iterations of updating cluster features
6      #       teu: the temperture of distance matrix
7      # output:
8      #       cluster_features: shape [B, hw, C]
9
10     B, C, H, W = features.shape
11
12     # initialize the cluster features
13     cluster_features = interpolate(features, cluster_features_shape)
14
15     # construct mask to constrain the interactions within local range
16     mask = calculate_mask(features.shape, cluster_features_shape)
17     mask = (~mask) * 1e16
18
19     features = features.reshape(B, C, -1).permute(0, 2, 1)  # (B, HW, C)
20     cluster_features = cluster_features.reshape(B, C, -1).permute(0, 2, 1)  # (B, hw, C)
21
22     for _ in range(num_iters):
23         # calculate L2 distance of features and cluster features, the shape distance_matrix is (B, hw,
        HW)
24         distance_matrix = L2_distance(features, cluster_features)
25         # mask remote distance through softmax
26         distance_matrix += mask
27         weights = (-distance_matrix / teu).softmax(dim=1)
28         # let the sum of weight of each cluster feature be 1
29         weights = weights / weights.sum(dim=2, keepdim=True).clamp_(min=1e-16)
30         cluster_features = matrix_product(weights, features)
31
32     return cluster_features
```

Listing 1: PyTorch example of token clustering layer.

```
1  def token_reconstruction_layer(cluster_features, features_before_clustering, features_after_clustering,
       k, teu):
2      # args:
3      #       cluster_features: shape [B, hw, C]
4      #       features_before_clustering: features of alpha-th layer before clustering, shape [B, hw, C]
5      #       features_after_clustering: features of alpha-th layer before clustering, shape [B, HW, C]
6      #       k: topk parameter
7      #       teu: the temperture of weight matrix
8      # output:
9      #       features: reconstruction features, shape [B, HW, C]
10
11     # calculate L2 distance between features and cluster_features
12     distance = L2_distance(features_before_clustering, features_after_clustering)
13     weight = exp(-teu * distance)
14     # only remain the k weight of the most simliar features, calculating mask
15     topk, indices = topk(weight, k=k, dim=2)
16     mink = min(topk, dim=-1).values
17     mink = mink.unsqueeze(-1).repeat(1, 1, weight.shape[-1])
18     mask = greater_or_equal(weight, mink)
19     weight = weight * mask
20
21     weight = weight / weight.sum(dim=2, keepdim=True).clamp_(min=1e-16)
22     features = matrix_product(weight, cluster_features)
23
24     return features
```

Listing 2: PyTorch example of token reconstruction layer.

Table 2 summarizes the hyper-parameters, including the inserted positions $\alpha$ & $\alpha+\beta$ of the window token clustering layer & window token reconstruction layer, the number of remaining transformer layers after the token reconstruction layer $\gamma$, the total number of transformer layers L, the number of window tokens before clustering K $\times$ K, the number of window tokens after clustering k $\times$ k, the number of neighboring pixels $\lambda$, the number of EM iterations $\kappa$, the temperature value $\tau$, and the number of nearest neighbors k, for Mask2Former and SwinV2-L + HTC++.

## C. More Evaluation Details

We illustrate the evaluation details used for measuring the GFLOPs and FPS of different methods in Table 3. We choose the input resolutions for different methods with different backbones according to their official implementations. To illustrate the effectiveness of our method more accurately, we do not include the complexity and latency brought by the especially heavy detection heads or segmentation heads within Mask2Former and SwinV2-L + HTC++. For example, the GFLOPs of

SwinV2-L backbone accounts for only $56.7\%$ of the whole model, therefore, we only report the GFLOPs and FPS improvements of our method over the backbone.

## D. Comparison with EViT [3] on Dense Prediction

To demonstrate the advantage of our approach over the representative method that is originally designed for the image classification tasks, i.e., EViT [3], we report the detailed comparison results in Figure 3. The original EViT propose to identify and only keep the top $\rho\%$ tokens according to their attention scores relative to the [class] token. Specifically, we follow the official implementations to insert the token identification module into the 8-th, 14-th, and 20-th layer of ViT-L/16 (with 24 layers in total) to decrease the number of tokens by (1-$\rho\%$), respectively. We report the results of EViT by choosing $\rho\%$=60%/70%/80%/90% in Figure 3. Accordingly, we can see that our method significantly outperforms EViT across various GFLOPs & FPS settings when evaluating without either re-training or fine-tuning.

The EViT can not be used for dense prediction directly, as it only keeps around $21.6\% \sim 72.9\%$ of the tokens at last. To reconstruct the missed token representations over the abandoned positions, we apply two different strategies, including (i) reusing the representations before the corresponding token identification module, and (ii) using our token reconstruction layer to reconstruct the missed token representations according to Figure 2a. We empirically find the first strategy achieves much worse results, thus choosing the second strategy by default.

## E. Adapting DynamicViT [6] for Dense Prediction

To adapt DynamicViT for dense prediction tasks, we propose to add multiple token reconstruction layers to reconstruct high-resolution representations from the selected low-resolution representations iteratively. Figure 2 (b) presents more details of the overall framework. We also report the comparison results in Table 4.

## F. Comparison with Clustered Attention [9], ACT [10], and SMRF [2]

We illustrate the key differences between our approach and the existing clustered attention approaches [9, 10, 2] the following two aspects: (i) These clustering attention methods perform clustering within each multi-head self-attention layer (MHSA) independently while our approach only performs clustering once with the token clustering layer and refines the clustered representations with the following transformer layers. Therefore, our approach introduces a much smaller additional overhead caused by the clustering operation. (ii) These clustering attention methods only reduce the computation cost of each MHSA layer equipped with clustering attention as they maintain the high-resolution representations outside the MHSA layers while Our approach can reduce the computation cost of both MHSA layers and feed-forward network (FFN) layers after the token clustering layer. We further summarize their detailed differences and the experimental comparison resultswith ACT [10](without retraining) in Table 5 and Table 6, respectively.

According to the results in Table 6, we can see that (i) ACT also achieves strong performance without retraining, (ii) our approach is a better choice considering the trade-off between performance and FPS & GFLOPs, e.g., our method achieves close performance as ACT (51.32 vs. 51.38) while running 70% faster (9.1 vs. 5.3) and saving more than $35\%$ GFLOPs (388.2 vs. 614.7).

## G. Visualization

We first present the visual comparison results of our approach in Figure 4a, which shows three different configurations over Segmenter+ViT-L/16 achieve 32.13%/48.21%/51.32% when setting the cluster size h × w as $8 \times 8/16 \times 16/24 \times 24$, respectively.

Then, we visualize both the original feature maps and the clustering feature maps in Figure 4b. Accordingly, we can see that the clustering feature maps, based on our token clustering layer, well maintain the overall structure information carried in the original high-resolution feature maps.

Last, to verify the redundancy in the tokens of vision transformer, we visualize the attention maps of neighboring tokens in Figure 5.

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

Table 1: Illustrating the hyper-parameter settings used for Segmenter, DPT, and SWAG.

| Method | Backbone | Dataset | $\alpha$ | $\alpha+\beta$ | $\gamma$ | L | $\frac{H}{P} \times \frac{W}{P}$ | h × w | $\lambda$ | $\kappa$ | $\tau$ | k |
|---|---|---|---|---|---|---|---|---|---|---|---|---|
| Segmenter [8] | ViT-L/16 | ADE20K | 10 | | | | 40 × 40 | 28 × 28 | | | | |
| | | Cityscapes | 12 | 24 | 0 | 24 | 48 × 48 | 32 × 32 | 5 × 5 | 5 | 50 | 20 |
| | | PASCAL-Context | 14 | | | | 30 × 30 | 15 × 15 | | | | |
| DPT [5] | R50+ViT-B/16 | KITTI | 2 | 12 | 0 | 12 | 76 × 22 | 28 × 28 | 5 × 5 | 5 | 5 | 20 |
| | | NYUv2 | 3 | | | | 40 × 30 | 16 × 16 | 7 × 7 | 5 | 10 | 50 |
| SWAG [7] | ViT-H/14 | ImageNet-1K | 8 | 32 | 0 | 32 | 37 × 37 | 25 × 25 | 7 × 7 | 5 | 1 | 20 |
| | ViT-L/16 | | 8 | 24 | 0 | 24 | 32 × 32 | 22 × 22 | 9 × 9 | 5 | 1 | 20 |

Table 2: Illustrating the hyper-parameter settings used for Mask2Former and SwinV2-L + HTC++.

| Method | Backbone | Dataset | $\alpha$ | $\alpha+\beta$ | $\gamma$ | L | K × K | k × k | $\lambda$ | $\kappa$ | $\tau$ | k |
|---|---|---|---|---|---|---|---|---|---|---|---|---|
| Mask2Former [1] | Swin-L | COCO (panoptic seg.) | 10 | | | | | | 7 × 7 | 5 | 20 | 10 |
| | | ADE20K (semantic seg.) | 8 | 22 | 2 | 24 | 12 × 12 | 8 × 8 | 5 × 5 | 5 | 100 | 10 |
| | | COCO (instance seg.) | 12 | | | | | | 11 × 11 | 5 | 100 | 60 |
| SwinV2-L + HTC++ [4] | SwinV2-L | COCO (object det.) | 12 | 22 | 2 | 24 | 32 × 32 | 23 × 23 | 5 × 5 | 5 | 33 | 20 |

Table 3: Illustrating the hyper-parameter settings used for measuring FPS and GFLOPs.

| Method | Backbone | with Head | Dataset | Input resolution |
|---|---|---|---|---|
| Segmenter [8] | ViT-L/16 | ✓ | ADE20K | 640 × 640 |
| | | | Cityscapes | 768 × 768 |
| | | | PASCAL-Context | 480 × 480 |
| DPT [5] | R50+ViT-B/16 | ✓ | KITTI | 1216 × 352 |
| | | | NYUv2 | 640 × 480 |
| SWAG [7] | ViT-H/14 | ✓ | ImageNet-1K | 518 × 518 |
| | ViT-L/16 | | | 512 × 512 |
| Mask2Former [1] | Swin-L | ✗ | COCO (panoptic seg.) | 1152 × 1152 |
| | | | ADE20K (semantic seg.) | 1152 × 1152 |
| | | | COCO (instance seg.) | 1152 × 1152 |
| SwinV2-L + HTC++ [4] | SwinV2-L | ✗ | COCO (object det.) | 1024 × 1024 |

Table 4: Comparison to parametric methods based on Segmenter [8].

| Dataset | Method | Parametric | Fine-Tuning | GFLOPs | mIoU |
|---|---|---|---|---|---|
| ADE20K | Dynamic ViT ($\rho = 0.7$) | ✓ | ✓ | 455.6 | 45.62 |
| | Dynamic ViT ($\rho = 0.8$) | ✓ | ✓ | 513.3 | 47.89 |
| | Dynamic ViT ($\rho = 0.9$) | ✓ | ✓ | 583.0 | 50.42 |
| | Ours (h × w = 16 × 16) | ✗ | ✗ | 315.1 | 48.21 |
| | Ours (h × w = 20 × 20) | ✗ | ✗ | 347.2 | 50.17 |
| | Ours (h × w = 24 × 24) | ✗ | ✗ | 388.2 | 51.32 |

Table 5: Illustrating the differences between clustered attention [9], ACT [10], SMRF [2], and our approach.

| Cluster method | query | key-value | FFN | #clustering layers |
|---|---|---|---|---|
| Clustered Attention [9] | ✓ | ✗ | ✗ | # MHSA layers |
| ACT [10] | ✓ | ✗ | ✗ | # MHSA layers |
| SMRF [2] | ✓ | ✓ | ✗ | # MHSA layers |
| Ours | ✓ | ✓ | ✓ | 1 |

Table 6: Comparison results with ACT [10].

| Cluster method | FPS | GFLOPs | mIoU |
|---|---|---|---|
| Segmenter+ViT-B/16 | 6.2 | 659.0 | 51.82 |
| Segmenter+ViT-B/16+Ours(h×w=24 × 24) | 9.1 | 388.2 | 51.32 |
| Segmenter+ViT-B/16+Ours(h×w=28 × 28) | 8.8 | 438.9 | 51.56 |
| Segmenter+ViT-B/16+ACT(#query-hashes=16) | 5.8 | 578.7 | 48.12 |
| Segmenter+ViT-B/16+ACT(#query-hashes=24) | 5.3 | 614.7 | 51.38 |
| Segmenter+ViT-B/16+ACT(#query-hashes=32) | 5.0 | 638.2 | 51.64 |

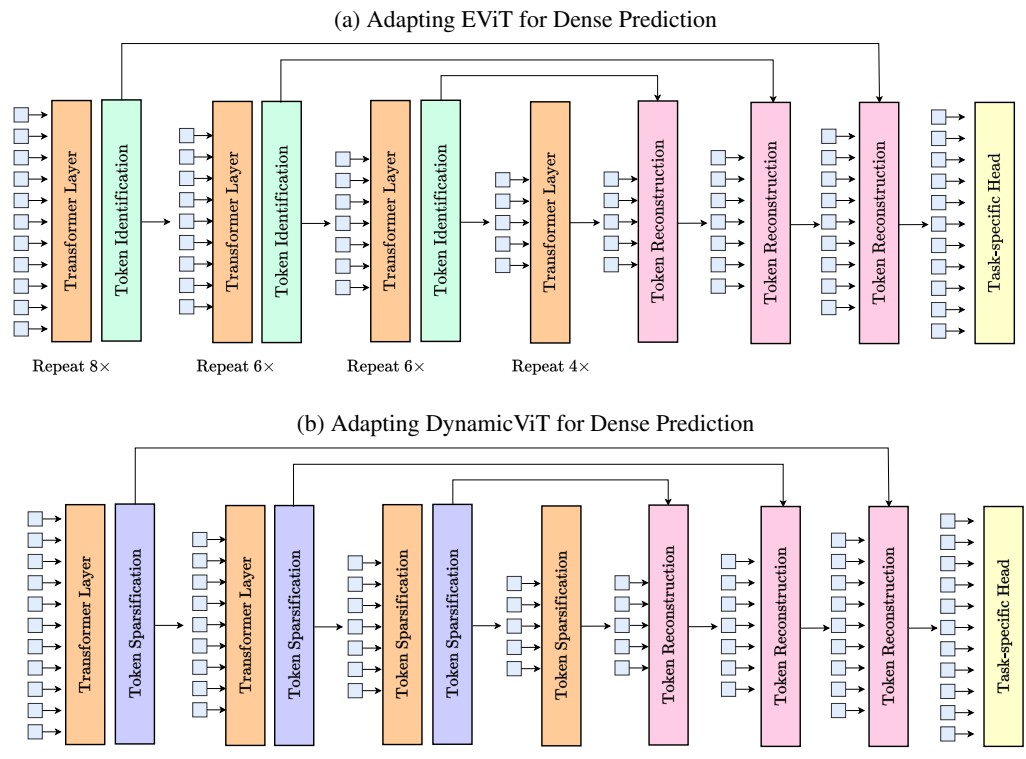

(a) Adapting EViT for Dense Prediction

(b) Adapting DynamicViT for Dense Prediction

Figure 2: Illustrating how to adapt the EViT [3] and DynamicViT [6] for dense prediction based on ViT-L/16 with 24 transformer layers. Following the proposed token reconstruction scheme, we estimate the semantic relations based on the representations before each token identification layer [3] or token sparsification layer [6].

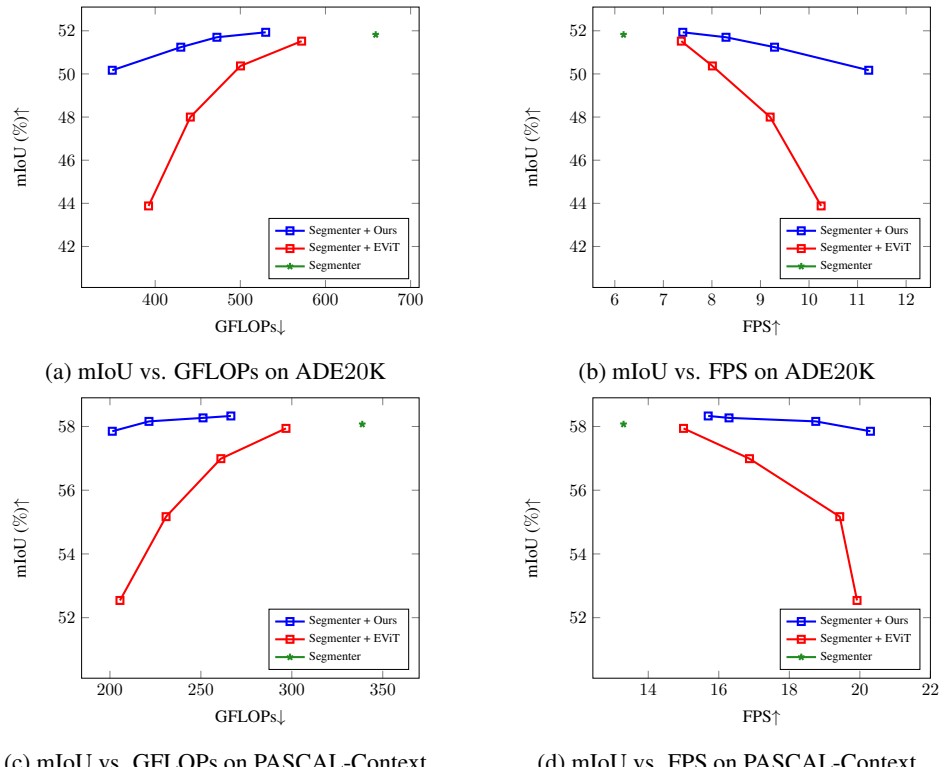

(a) mIoU vs. GFLOPs on ADE20K

(b) mIoU vs. FPS on ADE20K

(c) mIoU vs. GFLOPs on PASCAL-Context

(d) mIoU vs. FPS on PASCAL-Context

Figure 3: Comparison with EViT [3] on ADE20K and PASCAL-Context semantic segmentation task based on Segmenter with ViT-L/16. ↑ and ↓ represent higher is better and lower is better respectively.

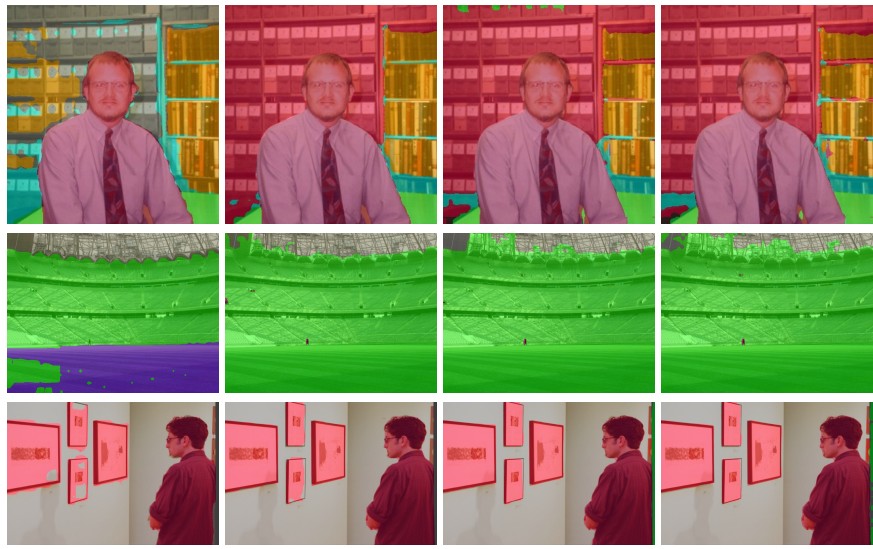

(a) ADE20K example segmentation results of our approach with h × w as $8 \times 8$, $16 \times 16$, $24 \times 24$ on the left three columns, respectively. The right-most column shows the results of the original Segmenter+ViT-L/16. We can see that our approach achieves consistently better segmentation results with increasing clustered output resolutions from left to right.

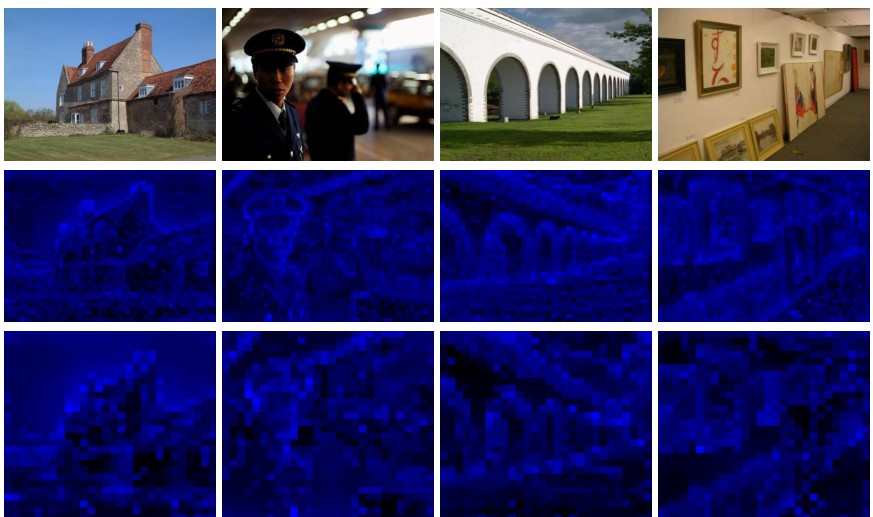

(b) ADE20K example visualization of the original feature maps (2-ed row) and the clustering feature maps (3-rd row). We can see that the clustering feature maps still maintain the structure information presented in the original high-resolution feature maps, thus showing the potential benefits of our token clustering scheme.

Figure 4: Visualizations of segmentation results in (a) and feature maps in (b). We choose Segmenter+ViT-L/16 on ADE20K to generate the above segmentation results and the feature map visualizations.

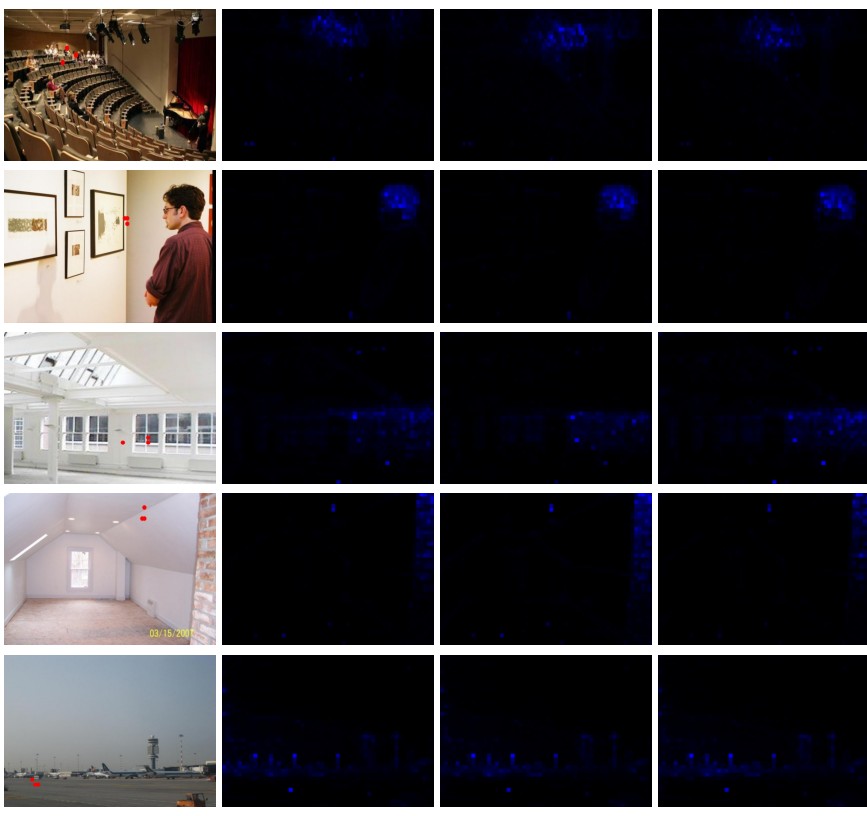

Figure 5: Visualizations of the attention maps of neighboring sampled positions. We mark the sampled positions with red point markers. We can see that the neighboring positions share highly similar attention maps, which matches the redundancy observation in the Adaptive Clustering Transformer (ACT) [10].