# OpenReview forum: "Expediting Large-Scale Vision Transformer for Dense Prediction without Fine-tuning"
_NeurIPS.cc/2022/Conference — NeurIPS 2022 Accept_

### Official Review · Reviewer_Ldum · 2022-07-10

**Rating:** 5
**Confidence:** 4
**Soundness:** 4 excellent
**Presentation:** 4 excellent
**Contribution:** 2 fair

**Summary:**

The authors observed that the presence of local spatial representation redundancy and high definition representation in the vision transformer contribute to dense prediction. Therefore, this paper proposes the token clustering layer and the token reconstruction layer to reduce computation and memory cost. The proposed modules are non-parametric operations and are applicable to different models and multiple dense prediction tasks.

**Questions:**

I suggest that authors provide visualizations of the original features and clustering features in the paper or in the appendix, if possible.

**Limitations:**

I do not see any potential negative societal impact for this work.

**Strengths And Weaknesses:**

Strengths:
1. The paper is clearly motivated, well written and with sufficient experimentation.
2. The proposed token clustering and reconstruction are non-parametric operations and therefore can be implemented directly on the well-trained model without additional fine-tuning. Taking existing trained models, the proposed method significantly improves the efficiency of the large vision transformer on the dense prediction task at a low cost in terms of precision.
3. The authors' observations are insightful, and the methods proposed are clever and easy to implement. In addition to the five dense prediction tasks in the paper, I think the proposed approach also has the potential to be extended to other vision transformer-based tasks.
4. The authors also provides the extension strategy and experiments for Swin-Transformer in addition to ViT.

Weaknesses:
1. Since the authors claimed that there is redundancy in the tokens of the vision transformer and that the proposed method is based on clustering, some visualizations could better represent this observation.
2. Clustering attention has been explored by previous approaches such as clustering attention[1] and adaptive clustering attention[2]. Specifically, ACT can slim the high-resolution transformer without retraining. The author is highly encouraged to discuss relationship between previous works.

Overall, I think the proposed approach is straightforward and effective. However, the paper lack comparison with previous similar approaches. The author is encouraged to discuss related works especially on clustering attention topic.
[1] Fast transformers with clustered attention
[2] End-to-end object detection with adaptive clustering transformer
[3] SMYRF: Efficient Attention using Asymmetric Clustering

---

> ### Author Response · Authors · 2022-08-02
> **Response to Reviewer Ldum**
>
> We thank the reviewer for the careful reviews and constructive suggestions. We answer the questions as follows.
>
> ___
> >"Since the authors claimed that there is redundancy in the tokens of the vision transformer and that the proposed method is based on clustering, some visualizations could better represent this observation."
>
> A: Thanks for your valuable suggestions. We follow your suggestions to visualize the attention maps of neighboring tokens in Figure 5 of the revised supplementary. We would like to revise the visualizations if you could provide any further detailed suggestions.
>
> ___
> >"Clustering attention has been explored by previous approaches such as clustering attention[1] and adaptive clustering attention[2]. Specifically, ACT can slim the high-resolution transformer without retraining. The author is highly encouraged to discuss relationships between previous works. Overall, I think the proposed approach is straightforward and effective. However, the paper lack comparison with previous similar approaches. The author is encouraged to discuss related works, especially on clustering attention topics. [1] Fast transformers with clustered attention [2] End-to-end object detection with adaptive clustering transformer [3] SMYRF: Efficient Attention using Asymmetric Clustering"
>
> A: Thanks for pointing out these GREAT works on clustering attention[1][2][3] and we will include the references and the following discussions in the revision.
>
> 👉 The key differences are:
> 1. These clustering attention methods perform clustering within each multi-head self-attention layer (MHSA) independently. **vs.** Our approach only performs clustering once with the token clustering layer and refines the clustered representations with the following transformer layers. Therefore, our approach introduces a much smaller additional overhead caused by the clustering operation.
> 2. These clustering attention methods ONLY reduce the computation cost of each MHSA layer equipped with clustering attention as they maintain the high-resolution representations outside the MHSA layers. **vs.** Our approach can reduce the computation cost of both MHSA layers and feed-forward network (FFN) layers after the token clustering layer.
>
> We further summarize their detailed differences in the following Table:
>
> | Cluster method  | Query | Key-Value | FFN | # Clutering times |
> | ----------- | ----------- | ----------- | ----------- | -----------:|
> | ClusteredAttention[1]  | ✔️ | ❌ | ❌ | # MHSA layers |
> | ACT[2] | ✔️ | ❌ | ❌ | # MHSA layers |
> | SMYRF[3] | ✔️ | ✔️  | ❌ | # MHSA layers |
> | Ours | ✔️  | ✔️  | ✔️ | $1$ |
>
> In the above Table, we use ✔️ to mark processing the clustered representations and ❌ to mark processing the original representations.
>
> 👉 The similarities are: ACT[2] and SMYRF[3] can also slim & accelerate vision transformer without retraining by applying locality-sensitive hashing (LSH). Specifically, ACT focused on accelerating DETRs while SMYRF focused on BERTs and BigGANs. We compare with the more representative ACT[2] as follows.
>
>
> 👉 Comparison with ACT[2] without retraining:
> We follow your suggestions to report the segmentation results of Segmenter+ViT-L based on the official implementations of ACT[2]:
>
> | Cluster method  | FPS | GFLOPs | mIoU |
> | -----------| ----------- | ----------- | -----------:|
> | Baseline  | $6.2$ | $659.0$ | $51.82$ |
> | Ours ($\mathrm{h}\times\mathrm{w}=24\times24$)  | $9.1$ | $388.2$ | $51.32$ |
> | Ours ($\mathrm{h}\times\mathrm{w}=28\times28$)  | $8.8$ | $438.9$ | $51.56$ |
> | ACT (#query-hashes=16)  | $5.8$ | $578.7$ | $48.12$ |
> | ACT (#query-hashes=24)  | $5.3$ | $614.7$ | $51.38$ |
> | ACT (#query-hashes=32)  | $5.0$ | $638.2$ | $51.64$ |
>
> According to the above results, we can see that (1) ACT also achieves strong performance without retraining, (2) our approach is a better choice considering the trade-off between performance and FPS & GFLOPs, e.g., our method achieves close performance as ACT (51.32 vs. 51.38) while running 70% faster (9.1 vs. 5.3) and saving more than 35% GFLOPs (388.2 vs. 614.7).
>
>
> ___
> >"I suggest that authors provide visualizations of the original features and clustering features in the paper or in the appendix, if possible."
>
> A: Thanks for your suggestions and we have included the related visualizations following your suggestions in Figure 4 (b) of the supplementary.

---

> > ### Comment · Reviewer_Ldum · 2022-08-08
> > **Good rebuttal**
> >
> > Thanks for your responses. The author is encouraged to add the rebuttal contents to the main paper in the future.

---

> > > ### Author Response · Authors · 2022-08-08
> > > **Thanks for the Response of Reviewer Ldum**
> > >
> > > We thank the reviewer for your careful response and for increasing the rating.
> > >
> > > We will add the rebuttal contents to the main paper in the revision following your valuable suggestions.

---

### Official Review · Reviewer_kZ4Z · 2022-07-11

**Rating:** 5
**Confidence:** 4
**Soundness:** 3 good
**Presentation:** 3 good
**Contribution:** 3 good

**Summary:**

This paper introduces two non-parametric operators to efficiently down-sampling some intermedia tokens to accelerate the large-scale vision transformers for dense prediction. One is a token clustering layer  to decrease the number of tokens and the other one is a token reconstruction layer to recover the number of tokens. These two operators need no training or finetuning, while the related experimental results prove the effectiveness of them in typical benchmarks.

**Questions:**

None.

**Ethics Review Area:**

["I don’t know"]

**Limitations:**

See the above discussion about weaknesses.

**Strengths And Weaknesses:**

Strengths:
The motivation of the paper is clear and solid. While the current existing works for efficient transfomer need training or finetuning, this work introduces two non-parametric operators and brings some meaningful insight for this topic.
The experimetal results are sufficient to support the proposed work.
The paper is well written and easy to follow.

Weaknesses:
From Table 2, I see that the proposed operators actually have limited performance when increasing the downsampling stride. I think this limitation is normal for non-paramettic modules, but I also think the author should also compare with the other parametric methods such as [47,52,55] here to give a comprehensive discussion.

Overall, I think this paper is technically sound and achieve good performance. Thus I vote for borderline accept.

---

> ### Author Response · Authors · 2022-08-02
> **Response to Reviewer kZ4Z**
>
> We thank the reviewer for the careful reviews and constructive suggestions. We answer the questions as follows.
>
> ___
> >"From Table 2, I see that the proposed operators actually have limited performance when increasing the downsampling stride. I think this limitation is normal for non-parametric modules, but I also think the author should also compare with the other parametric methods such as [47,52,55] here to give a comprehensive discussion."
>
> A:
> 👉 Comparison with TokenPooling[47]/DynamicViT[52]/TokenLearner[55]:
> 1. [52] and [55] are **parametric** that introduce additional parameters while [47] and our approach are **non-parametric**. For example, [52] introduces a trainable prediction module to estimate the importance score of each token, [55] uses a learnable convolution to predict a group of spatial attention maps and generates a set of token vectors accordingly.
> 2. [47] and [52] focus on accelerating vision transformers for only image classification tasks. **They can not be applied for dense prediction tasks directly as they only keep a small set of selected/clustering tokens, thus losing the spatial information that is necessary for dense prediction tasks**. Our approach maintains the spatial information and can be used to **accelerate both image classification and various dense prediction tasks**, especially segmentation, object detection, and depth estimation.
> 3. [47] and [52] only reduce the number of tokens while [55] and our approach further propose to increase the number of tokens with either TokenFuser or token reconstruction layer, where the difference is that [55] repeatedly applies the combination of TokenLearner & TokenFuser before/after each transformer layer while our approach only applies token clustering layer & reconstruction layer once through the whole transformer. Therefore, our approach introduces a much smaller additional overhead caused by the clustering & reconstruction operations when compared to [55].
>
>
> 👉 Comparison experiments: considering both [47] and [52] can not be used for dense prediction tasks directly, we **apply our token reconstruction layer to adapt the most representative [52] for segmentation tasks**. We illustrate more details in Figure 2 (b) of the supplementary and report the comparison results as follows:
>
> | Cluster method  | GFLOPs | Parametric | Fine-tuning | mIoU |
> | ----------- | ----------- | ----------- | ----------- | -----------:|
> | DynamicViT($\rho=0.7$) | $455.6$ | ✔️ | ✔️ | $43.79$ |
> | DynamicViT($\rho=0.8$) | $513.3$ | ✔️ | ✔️ | $46.78$ |
> | DynamicViT($\rho=0.9$) | $583.0$ | ✔️ | ✔️ | $46.95$ |
> | Ours ($\mathrm{h}\times\mathrm{w}=8\times8$)  | $274.0$ | ❌ | ❌ | $32.13$ |
> | Ours ($\mathrm{h}\times\mathrm{w}=16\times16$) | $315.1$ | ❌ | ❌ | $48.21$ |
> | Ours ($\mathrm{h}\times\mathrm{w}=24\times24$) | $388.2$ | ❌ | ❌ | $51.32$ |
>
> According to the above results, we can see that our approach consistently outperforms DynamicViT with more accurate segmentation results while requiring much fewer GFLOPs.

---

> ### Author Response · Authors · 2022-08-09
> **Looking forward to hearing the response from Reviewer kZ4Z**
>
> We thank the reviewer for the previous careful reviews and valuable suggestions.
>
> We have learned a lot through the suggested comparisons with TokenPooling[47]/DynamicViT[52]/TokenLearner[55]. We also hope to learn more from your further valuable suggestions.

---

### Official Review · Reviewer_sJzo · 2022-07-12

**Rating:** 5
**Confidence:** 4
**Soundness:** 2 fair
**Presentation:** 2 fair
**Contribution:** 2 fair

**Summary:**

When using the Transformer for dense image prediction, detection, and segmentation, the large number of tokens can impose a heavy computational burden. Therefore, the core of this paper is to propose clustering tokens to obtain a low-resolution representation to reduce the computational burden, and to perform attention on the low-resolution token cluster, followed by a token reconstruction layer to reconstruct the high-resolution token representation. The authors' experiments verify that the proposed approach significantly improves the FPS and reduces the GFLOPS index for segmentation, detection, and depth estimation tasks when using Transformer with only a small loss of AP.

**Questions:**

The use of the superpixel method, while reducing the size of the token, must lead to a loss of semantic information. And after the reconstruction of the layers, there must be noise in the regained token. My question is how the loss of semantic information in each layer compares to that of the original ViT after using the two modules proposed by the authors

**Limitations:**

I personally believe that the work in this article has contributed to reducing the computational burden of the Transformer. However, the two modules proposed in the article should be further verified and reasoned in detail to show the reader more details.

**Strengths And Weaknesses:**

Strengths:
This paper introduces the superpixel scheme into the Transformer structure, which really reduces the computational burden on the Transformer. The author's solution ensures that the entire computational process is differentiable and is a practical solution.

Weaknesses
The Token Clustering Layer in the paper is similar to the algorithm cited in [30], where the center of the superpixel is obtained and then iterated until the result of the latest clustering is obtained. Although better results are obtained, the method is only introduced in and the innovation point is not sufficient. And I think the author's description of the two modules is not detailed enough and should be further illustrated with pictures. The experimental part should be more experimental on multiple datasets for each task.

---

> ### Author Response · Authors · 2022-08-02
> **Response to Reviewer sJzo**
>
> We thank the reviewer for the careful reviews and constructive suggestions. We answer the questions as follows.
>
> ___
> >"The Token Clustering Layer in the paper is similar to the algorithm cited in [30], where the center of the superpixel is obtained and then iterated until the result of the latest clustering is obtained. Although better results are obtained, the method is only introduced and the innovation point is not sufficient. And I think the author's description of the two modules is not detailed enough and should be further illustrated with pictures. The experimental part should be more experimental on multiple datasets for each task."
>
> A: Thanks for your constructive comments!
>
> 👉**Concerns about the innovation**: we agree that the token clustering layer is similar to [30]. We would like to stress the importance of our token reconstruction layer and we show that it can be used to adapt the very recent EViT[1] and DynamicViT[2] for dense prediction tasks in the supplementary. Please refer to the **novelty & contribution** in the general response.
>
> 👉**Concerns about the details**: we have followed your suggestions and illustrated the details of the two modules in Figure 1 in the revised supplementary. Besides, we also provide the PyTorch example implementations in the supplementary to clarify more details.
>
> 👉**Concerns about the experiments**: we have already reported the object detection results on COCO, segmentation results on ADE20K/PASCAL-Context/Cityscapes, and monocular depth estimation results on KITTI/NYUv2. We would like to improve the experiments if you could provide more detailed suggestions.
>
> [1] Liang, Youwei, et al. "EViT: Expediting Vision Transformers via Token Reorganizations." ICLR 2022.
>
> [2] Rao, Yongming, et al. "Efficient Vision Transformers and CNNs with Dynamic Spatial Sparsification." NeurIPS 2021
>
> ___
> >"The use of the superpixel method, while reducing the size of the token, must lead to a loss of semantic information. And after the reconstruction of the layers, there must be noise in the regained token. My question is how the loss of semantic information in each layer compares to that of the original ViT after using the two modules proposed by the authors."
>
> A: We take Segmenter + ViT-L (on ADE$20$K, $\alpha=10$) as an example and analyze the loss of semantic information by calculating the cosine similarity between the reconstructed high-resolution feature $\mathbf{Z_{\alpha+\beta}}$ and the original high-resolution feature $\mathbf{Z^{original}_{\alpha+\beta}}$:
>
>
> | $\alpha+\beta$  | $12$ | $14$ | $16$ | $18$ | $20$ |$22$ |$24$ |
> | ----------- | ----------- | ----------- | ----------- | ----------- | ----------- | ----------- | -----------:|
> | $cos(\mathbf{Z_{\alpha+\beta}}, \mathbf{Z^{original}_{\alpha+\beta}})$ | $0.94$ | $0.95$ | $0.96$ | $0.96$ | $0.96$ | $0.96$ | $0.96$ | $0.96$ |
>
> Accordingly, we can see that our approach well maintains the semantic information and suffers less from the noise during the reconstruction process.
>
>
> ___
> >"I personally believe that the work in this article has contributed to reducing the computational burden of the Transformer. However, the two modules proposed in the article should be further verified and reasoned in detail to show the reader more details."
>
> A: We have included more details in the supplementary according to your suggestions as follows:
>
> 1. Figure 1 presents the detailed pipeline of our approach.
> 2. Listing 1 & 2 present the example implementations based on PyTorch.
>
> We would like to further improve the details if you have any further advice.

---

> ### Author Response · Authors · 2022-08-09
> **Looking forward to hearing the response from Reviewer sJzo**
>
> We thank the reviewer for the previous careful reviews and valuable suggestions.
>
> We have learned a lot from the response from the other reviewers. We also hope to learn more from your further suggestions.

---

### Official Review · Reviewer_Wt6a · 2022-07-26

**Rating:** 5
**Confidence:** 1
**Soundness:** 2 fair
**Presentation:** 3 good
**Contribution:** 3 good

**Summary:**

This paper accelerates vision transformers for dense prediction without finetuning. This is done by (i) using the token clustering layer to cluster the neighboring tokens; (2) using the token reconstruction layer to re-create the high-resolution representations.   The result of proposed method achieves state-of-the-art performance on five dense prediction tasks

**Questions:**

According to my understanding, author speed up Transformer by reducing token number and restore feature representation from these clustered tokens.  However, I do not understand why this approach could avoid finetune.

**Limitations:**

Could authors provide detailed discussion for reason why their acceleration method can avoid finetune.

**Strengths And Weaknesses:**

This paper proposes two layers (token clustering layer and token reconstruction layer) and apply them into Swin Transformer. The two ideas are simple and easy to follow. The setting of experiments is clear and the number of experiments is adequate.


This paper fulfills with texts and numbers. I think that authors should provide visual comparison for  detection, segmentation and depth estimation

---

> ### Author Response · Authors · 2022-08-02
> **Response to Reviewer Wt6a**
>
> We thank the reviewer for the careful reviews and constructive suggestions. We answer the questions as follows.
>
>
> >"This paper fulfills with texts and numbers. I think that authors should provide a visual comparison for detection, segmentation, and depth estimation"
>
> A: Good points! We have followed your suggestions and included some visual comparison results, from the ADE$20$K segmentation benchmark, in the revised supplementary. Specifically, Figure 4 (a) presents the visual comparisons of our approach under different settings of cluster size on the ADE$20$K semantic segmentation task. We would like to include more visual comparisons on both detection and depth estimation benchmarks in the final revision.
>
> ___
> >"According to my understanding, the author speeds up Transformer by reducing the token number and restoring feature representation from these clustered tokens. However, I do not understand why this approach could avoid finetuning. Could authors provide detailed discussion for the reason why their acceleration method can avoid finetune?"
>
> A: The reasons include the following two aspects:
>
> 1. Our token clustering/reconstruction layers are **non-parametric**, thus avoiding retraining any additional parameters.
> 2. The reconstructed high-resolution representations **maintain high semantic similarity** with the original high-resolution representations.
>
> We take Segmenter + ViT-L (on ADE$20$K, $\alpha$=10) as an example and analyze the semantic similarity between the reconstructed high-resolution feature $\mathbf{Z_{\alpha+\beta}}$ (with our approach) and the original high-resolution feature $\mathbf{Z^{original}_{\alpha+\beta}}$ (with original ViT-L):
>
> | $\alpha+\beta$  | $12$ | $14$ | $16$ | $18$ | $20$ |$22$ |$24$ |
> | ----------- | ----------- | ----------- | ----------- | ----------- | ----------- | ----------- | -----------:|
> | $cos(\mathbf{Z_{\alpha+\beta}}, \mathbf{Z^{original}_{\alpha+\beta}})$ | $0.94$ | $0.95$ | $0.96$ | $0.96$ | $0.96$ | $0.96$ | $0.96$ | $0.96$ |
>
> In the above Table, $\alpha$ represents the inserted layer index of our token clustering layer, and $\alpha+\beta$ represents the inserted layer index of our token reconstruction layer. Accordingly, we can see that the cosine similarities are consistently high across different transformer layers between the reconstructed high-resolution feature $\mathbf{Z_{\alpha+\beta}}$ (with our approach) and the original high-resolution feature $\mathbf{Z^{original}_{\alpha+\beta}}$. In other words, **our approach well maintains the semantic information carried in the original high-resolution feature maps and thus is capable of avoiding finetuning.**

---

> ### Author Response · Authors · 2022-08-09
> **Looking forward to hearing the response from Reviewer Wt6a**
>
> We thank the reviewer for the previous careful reviews and constructive suggestions.
>
> We have learned a lot from the response from the other reviewers. We also would like to hear your further suggestions.

---

### Author Response · Authors · 2022-08-02
**General Response**

We thank all the reviewers for the careful reviews and constructive suggestions. We acknowledge the positive comments such as "the setting of experiments is clear and the number of experiments is adequate" (Reviewer Wt6a), "this article has contributed to reducing the computational burden" (Reviewer sJzo), "this paper is technically sound and achieves good performance" (Reviewer kZ4Z), and "clearly motivated, well written and with sufficient experimentation (Reviewer Ldum)".

Above all, we clarify the concerns from the following aspects:


> Motivation:

Our work aims to accelerate various advanced **SOTA large-scale vision transformers for dense prediction tasks without any additional re-training or fine-tuning** as they tend to be very expensive and require a lot of computation cost. Most of our experiments can be finished with **only 1$\times$ 16G V100 GPU**. However, if we need to re-train or fine-tune these large-scale vision transformers such as ViT-H and SwinV2-L, we will need to access at least **8$\times$ 32G V100 GPUs** considering their expensive training computation cost and huge GPU requirement. For example, even only fine-tuning SwinV2-L + HTC++ on COCO for $5$-epochs requires more than **240$\times$ GPU hours** ($30$ hours with $8{\times}$ 32G V100 GPUs).

> Importance:

**Large-scale vision transformer** models become increasingly important for dense prediction tasks and fine-tuning large-scale models becomes more and more expensive and impractical. Therefore, we hope our work could inspire more research efforts into exploring how to **accelerate large-scale vision transformers for dense prediction tasks without any additional re-training or fine-tuning** while maintaining the performance as much as possible.

> Novelty & Contribution:

👉The novelty of our work lies in two aspects:

(1) We are the **first to study how to accelerate SOTA large-scale vision transformers for dense prediction tasks without fine-tuning** (e.g., "Mask2Former + Swin-L" or "SwinV2-L + HTC++"). Besides, our approach also achieves much better accuracy and speedup trade-off when compared to the very recent ACT [1] that is based on a clustering attention scheme (Reviewer Ldum);

(2) Our **token clustering & reconstruction layers are capable of maintaining the semantic information encoded in the original high-resolution representations**. This is the very most important factor to avoid fine-tuning.


👉The key contribution of our work is in **designing an effective combination of a token clustering function $f(\cdot)$ and a token reconstruction function $g(\cdot)$, which aims to maximize the cosine similarity between the reconstructed high-resolution feature maps and the original ones without fine-tuning**:

$$\max_{f,g}\space cos(\mathcal{T}(\mathbf{Z_\alpha}), g(\mathcal{T}(f(\mathbf{Z_\alpha})))),$$

where $\alpha$ represents the inserted layer index of our token clustering layer, $\mathcal{T}(\mathbf{Z_\alpha})$ and $g(\mathcal{T}(f(\mathbf{Z_\alpha})))$ represent the original and reconstructed high-resolution feature maps, respectively. We implement $f(\cdot)$ and $g(\cdot)$ with the token clustering layer and the token reconstruction layer, respectively. $\mathcal{T}(\cdot)$ represents the combination of transformer layers between the token clustering layer and the token reconstruction layer.

👉**The design of our token reconstruction layer is the key and not straightforward essentially. We also show that our token reconstruction layer can be used to adapt the very recent EViT[2] and DynamicViT[3] for dense prediction tasks in the supplementary.**

[1] Zheng, Minghang, et al. "End-to-end object detection with adaptive clustering transformer." BMVC 2021.

[2] Liang, Youwei, et al. "EViT: Expediting Vision Transformers via Token Reorganizations." ICLR 2022.

[3] Rao, Yongming, et al. "Efficient Vision Transformers and CNNs with Dynamic Spatial Sparsification." NeurIPS 2021.

> Details & Visualizations:

We have **revised the supplementary material** to include (i) the details of our proposed two modules in Figure 1 and Listing 1 & 2, (ii) the details of how to adapt DynamicViT for dense prediction tasks in Figure 2 (b), (iii)  rich visualizations of both segmentation predictions and feature maps in Figure 4, and (iv) attention maps associated different sampled local neighboring positions in Figure 5. **We mark the modifications with blue-colored text**.

---

### Meta-Review · Area_Chair_Bmk5 · 2022-08-25

**Recommendation:** Accept
**Confidence:** Less certain

**Metareview:**

This paper presents a method to reduce the computational cost of a trained vision transformer for dense prediction. According to the authors' presented experiments, the method can accelerate the transformers effectively without retraining. Although some experiments are not throughout (as discussed below), I see potential in this method and would give the research community a try to see whether the method can be further generalized to other architectures.

The AC does see some strange experimental setups. For instance, it is strange that the authors chose Mask2Former to conduct experiments but uses Segmentor to conduct experiments on ADE20K. Mask2Former already provides quite a strong model on ADE20K. Why use Segmentor for ADE20K experiments? The AC also observe that the authors compare with ACT on Segmentor as well.

The code is strongly encouraged to release for letting the general public test the method on other architectures.

**Award:**

No

---

### Decision · Program_Chairs · 2022-09-14

Accept